# Variation of Autonomic Nervous System Function by Age and Gender in Thai Ischemic Stroke Patients

**DOI:** 10.3390/brainsci11030380

**Published:** 2021-03-17

**Authors:** Warawoot Chuangchai, Wiraporn Pothisiri, Phumdecha Chanbenjapipu

**Affiliations:** 1College of Population Studies, Chulalongkorn University, Bangkok 10330, Thailand; warawoot.c@chula.ac.th; 2Faculty of Nursing, Siam University, Bangkok 10160, Thailand; phumdecha.cha@siam.edu

**Keywords:** autonomic nervous system, ischemic stroke, heart rate variability, pulse transit time

## Abstract

Background: Ischemic stroke is one of the major causes of disability and mortality. Its effects on the autonomic nervous system (ANS) through nonlinear heart rate variability (HRV) and pulse transit time (PTT) have not been well explored among Thai patients. Objective: This study aims to demonstrate the association between ANS and ischemic stroke through nonlinear HRV and PTT. Methods: In total, 111 patients were enrolled in the study and their short-term HRV and PTT data were collected. Results: Parasympathetic tone was higher in elderly patients (≥60 years). The elderly patients had a higher SD1 but lower SD2 and SD2/SD1 than the younger patients, and a similar pattern was found in the female patients compared to the male patients. These findings were supported by the results of the Poincaré plots. Older and female patients had circular plots and approximately round plots, respectively. Moreover, the parasympathetic nervous system (PNS) response was moderate and positively associated with SD1 (*r* = 0.47, *p* < 0.001) and PTT (*r* = 0.29, *p* = 0.002), and negatively associated with SD2 and SD2/SD1 (*r* = −0.47, *p* < 0.001), after controlling for age and sex. Conclusions: The PNS response was predominant in older and female patients whereas the sympathetic response was lower than in the younger and male patients, which reflected certain characteristics of ANS response to ischemic stroke. Moreover, nonlinear parameters of SD1, SD2, SD2/SD1, and Poincaré plots including PTT are useful and recommended in investigating ANS, particularly in PNS, among ischemic stroke patients.

## 1. Introduction

Ischemic stroke is the most common type of stroke across the globe and is one of the serious causes of disability and mortality in Thailand [1]. Aging is the greatest risk factor, followed by male gender, within the Thai population [2]. An ischemic stroke is a neurovascular event characterized by the interruption of the blood supply within the brain [3]. It is closely associated with autonomic nervous system (ANS) dysfunction and cardiovascular responses, which likely result in the development of severe ischemic events [4].

Both the parasympathetic nervous system (PNS) and sympathetic nervous system (SNS) are branches of the ANS, which controls almost all visceral, vascular, and metabolic functions [3]. Heart rate variability (HRV) data have been demonstrated to reflect the activity of the ANS. HRV is a noninvasive measurement recorded via an electrocardiogram (ECG). HRV reflects variations between consecutive inter-beat-intervals as R-R (RR) intervals [5]. In terms of the time-domain parameters, previous studies have indicated the mean RR interval and the root mean square of successive RR interval differences (RMSSD) to positively associate with the PNS response. Meanwhile, the mean heart rate (HR) and Baevsky’s stress index have been shown to link with the SNS response [6,7]. Low frequency (LF), high frequency (HF), and the ratio of LF to HF power (LF/HF) are broadly used as the frequency domain parameters. Previous studies have indicated that LF and HF reflect SNS and PNS responses, respectively, whereas the LF/HF ratio reflects the sympathovagal balance [8,9].

In addition, nonlinear HRV parameters are determined by complex interactions among hemodynamic, electrophysiological, and humoral variables, as well as by autonomic and central nervous system regulations [10]. The parameters of the standard deviation (SD) SD1, SD2, and SD2/SD1 have been used for assessing ANS modulation. Previous studies have indicated that SD1 was likely linked to the PNS, whereas SD2 and SD2/SD1 were likely linked to the SNS [8,11]. Moreover, pulse transit time (PTT) is a measurement of duration between pulse peaks obtained by photoplethysmography (PPG) and ECG sensors. A previous study indicated that PTT was associated with arterial stiffness and blood pressure, which reflected vascular tone [12]. However, the nonlinear parameters and PTT in ischemic stroke patients have produced uncertain results and remain to be clarified.

Within the Thai population, the study of ischemic stroke using HRV data, particularly nonlinear HRV, is limited. Only a few studies indicated nonlinear parameters [13,14,15]. Recently, a study of HRV in chronic ischemic stroke patients was carried out with a small sample of 13 patients [16]. Numerous HRV studies have compared ischemic stroke patients with non-patients, exploring whether this association varies by age and sex [17,18,19,20]. There has been no study that has investigated the associations between ANS and age and sex among ischemic stroke patients. If the negative effects of ischemic stroke are to be reduced, then the underlying mechanisms of functional impairment need to be investigated. Therefore, this study aimed to investigate the ANS response and its association with age and sex among ischemic stroke patients. Nonlinear HRV parameters, including SD1, SD2, SD2/SD1, and Poincaré plots including PTT, were used. The present study will provide results which help to better understand ANS changes in patients with ischemic stroke.

## 2. Materials and Methods

### 2.1. Participants

In this cross-sectional study, patients who admitted as a new case with the first-ever stroke were observed from the stroke unit of Thammasat University Hospital (TUH), Thailand. All potentially eligible patients were screened using magnetic resonance imaging (MRI). Inclusion criteria were sex (both males and females), age (18 years or older), a diagnosis of ischemic stroke, and an early mobilization within 24 h of stroke onset. Exclusion criteria were a diagnosis with hemorrhagic stroke, valvular heart disease, or atrial fibrillation. 

Among them, three patients were unqualified to enroll in the study. As a result, a total of 111 ischemic stroke patients were included at the final stage. They ranged in age from 23 to 85 years and included 63 males (30–83 years) and 48 females (23–85 years), as shown in Figure 1. Moreover, all patients in the study were diagnosed with right middle cerebral artery (R-MCA) infarction, except one that had left MCA infarction. No insular lesions were detected.

### 2.2. Measurements

HRV data were collected during the acute phase with a short-term, 5-min ECG recording lead II protocol with fingertip PPG from resting patients, breathing normally, in a supine position. The parameters obtained for the HRV analysis included mean RR, mean HR, RMSSD, stress index, LF, HF, LF/HF ratio, SD1, SD2, and SD2/SD1, whereas the PPG data were collected for the PTT analysis. The ECG measurements were performed in the morning between 6 and 7 am. The room was maintained at a temperature of 25 °C. The ECG measurements were taken with a sampling rate of 1000 Hz using the PowerLab 26T data acquisition system (ADInstruments, Sydney, Australia) and recorded with LabChart software. In addition, blood pressure (BP) was assessed using an automatic blood pressure monitor (IntelliVue MP40, Philips, Amsterdam, The Netherlands). Systolic blood pressure (SBP) and diastolic blood pressure (DBP) were collected for pulse pressure (PP) and mean arterial pressure (MAP) [21].

### 2.3. Statistical Analysis

All HRV data were analyzed using Kubios HRV Standard software [22], version 3.3.1. Mean RR, RMSSD, and SD1 parameters were computed for the PNS index, whereas mean HR, stress index, and SD2 parameters were computed for the SNS. Levels of parasympathetic and sympathetic tones were categorized as low (lower than −2.01), relatively low (−2.00 to −1.01), average (−1.00 to 1.00), relatively high (1.01 to 2.00), and high (higher than 2.01). LF, HF and LF/HF ratio were computed in normalized unit (n.u.). Besides, the PTT was calculated based on time differences between peaks of PPG waves and R waves: PPG peak−R peak/1000. The PP was computed as SBP−DBP, and the MAP was obtained as DBP + 1/3 of PP. A normality test (Shapiro–Wilk) was applied to all collected values. Results were presented as mean ± standard deviation, number (%), or median and interquartile range (25–75°). Differences between groups were compared with an independent (unpaired) *t*-test for normal distribution and the Mann–Whitney *U* test for non-normal distribution. Relationships were analyzed with the nonparametric partial correlation while controlling for age and sex. A level of significance was set at *p* < 0.05 (two-tailed). All statistical analyses were performed with IBM SPSS Statistics version 22. Effect sizes were estimated and interpreted with Cohen’s *d*, eta-squared (η^2^), or correlation (*r*) [23].

## 3. Results

The present study included patients aged 61.57 ± 12.64 years, with 63 males (56.80%) and 48 females (43.20%). The PNS and SNS indices were 0.18 ± 2.00 and 1.36 ± 1.91, respectively. LF was lower than HF with the ratio of 1.45 ± 1.79. Additionally, SD1 was lower than SD2 with an SD2/SD1 of 1.47 ± 0.62. PTT, SBP, DBP, PP and MAP values were 0.24 ± 0.03, 149.01 ± 20.03, 82.95 ± 13.40, 66.05 ± 17.46, and 104.97 ± 13.63, respectively, as shown in Table 1.

The sample patients were categorized by age as under or over 60 years, with 41 young patients aged 48.46 ± 8.59 years (24 males aged 49.17 ± 7.60 years and 17 females aged 47.47 ± 9.98 years), and 70 elderly patients aged 69.24 ± 7.07 years (39 males aged 70.00 ± 7.30 years and 31 females aged 68.29 ± 6.77 years). In addition, patients were categorized by sex, with 63 male patients aged 62.06 ± 12.58 years and 48 female patients aged 60.92 ± 12.82 years.

Moreover, the young patients were associated with PNS index of −0.26 ± 1.45 and SNS index of 1.57 ± 1.97, while the older patients were associated with PNS index of 0.44 ± 2.22 and SNS index of 1.24 ± 1.87. Besides, the male patients were associated with PNS index of 0.07 ± 1.86 and SNS index of 1.45 ± 1.94, while the female patients were associated with PNS index of 0.33 ± 2.17 and SNS index of 1.25 ± 1.89.

By comparing the young and older patients as shown in Table 2, there were statistically significant differences between the groups by age (*p* < 0.001), PP (*p* = 0.004), SD2/SD1 (*p* = 0.014), SD2 (*p* = 0.023), SD1 (*p* = 0.024), parasympathetic tone (*p* = 0.030), and DBP (*p* = 0.041). In addition, there was a large effect by age (η^2^ = 0.69), relatively large effects in SD2 (*d* = 0.45) and SD1 (*d* = −0.45), medium effects in PP (η^2^ = 0.08) and SD2/SD1 (η^2^ = 0.06), and small effects in parasympathetic tone (η^2^ = 0.04) and DBP (η^2^ = 0.04). In contrast, no significant differences were found between the groups in sympathetic tone, LF, HF, LF/HF ratio, PTT, SBP, and MAP.

By comparing the male and female patients as shown in Table 2, statistically significant differences were found between the groups in SD2/SD1 (*p* = 0.014), SD2 (*p* = 0.023), SD1 (*p* = 0.026), and LF (*p* = 0.048). Moreover, relatively large effects were found in SD2 (*d* = 0.44), SD1 (*d* = −0.43), and LF (*d* = 0.38), and a small effect in SD2/SD1 (η^2^ = 0.05). In contrast, no significant differences were found between the groups in terms of age, autonomic tones, HF, LF/HF ratio, and PTT and BP indices.

As shown in Figure 2, patients under 60 years old mostly showed elliptical plots where the length of the SD1 line was shorter than the SD2 line. By contrast, patients over 60 years old mostly showed circular plots where the SD1 line was slightly shorter than the SD2 line. Male patients mostly showed oval-shaped plots, similar to patients under 60 years old. Conversely, female patients mostly showed approximately round plots where the SD1 line was nearly equal to the SD2 line.

After controlling for age and sex as shown in Table 3, there was statistical significance at *p* < 0.001 with moderate and negative partial correlations between PNS and SD2 (*r* = −0.47), SD2/SD1 (*r* = −0.47), LF (*r* = −0.38), and LF/HF ratio (*r* = −0.37). There was also a weak and negative partial correlation between PNS and DBP (*r* = −0.24, *p* = 0.012). There were moderate and positive partial correlations between PNS and SD1 (*r* = 0.47, *p* < 0.001), HF (*r* = 0.38, *p* < 0.001), and PTT (*r* = 0.29, *p* = 0.002). In addition, there was statistical significance at *p* < 0.001 with moderate and negative partial correlations between SNS and SD1 (*r* = −0.39) and HF (*r* = −0.28). There was also a weak and negative partial correlation between SNS and PTT (*r* = −0.21, *p* = 0.026). There were moderate and positive partial correlations between SNS and SD2/SD1 (*r* = 0.40, *p* < 0.001), SD2 (*r* = 0.39, *p* < 0.001), LF/HF ratio (*r* = 0.29 *p* = 0.002), DBP (*r* = 0.28 *p* = 0.003), and LF (*r* = 0.27, *p* = 0.004). In contrast, there was no statistical significance between autonomic indices and SBP, PP, and MAP.

## 4. Discussion

The present study aimed to explore the association between ANS and ischemic stroke through nonlinear HRV and PTT. The results indicated that ANS modulation among ischemic stroke patients differed in age and sex. Note that as there was no control group, this difference could simply be related to physiological differences in age and sex. Our results showed that older patients had a higher parasympathetic tone and PP, but a lower DBP than their younger counterparts. In addition, the SD1, SD2, and SD2/SD1 parameters differed between the younger and the older groups, and between men and women. Moreover, elliptical and circular Poincaré plot shapes characterized the young patients and older patients, respectively, while oval and approximately rounded plot shapes characterized the male patients and female patients, respectively. Furthermore, the PNS index was positively associated with HF, SD1 and PTT (negatively associated with the SNS), but negatively associated with LF, LF/HF ratio, SD2, SD2/SD1 and DBP (positively associated with the SNS).

The older patients had a higher parasympathetic tone than the younger patients. This result indicated that the ANS of older patients was more active within the PNS than in younger patients in balancing the SNS. This ANS dysfunction provided additional key information about ischemic stroke by suggesting that the difference between the elderly patients and the younger patients was not mainly involved with the low modulation of the SNS. It was rather due to an overactive PNS modulation. This could be explained by the DBP and PP results, which found that the elderly patients had a lower DBP and a higher PP than the younger patients. Regarding aging, low DBP caused by the loss of elasticity of arteries and high PP was affected by vasoconstriction. Together, these indicated that as blood flow to the heart and blood volume to the brain were lowered, then the PNS became more dominant.

The parameters of SD1, SD2, and SD2/SD1 were found to be useful indices in differentiating between the patients. The older patients had a higher SD1, but lower SD2 and SD2/SD1 than the younger patients; the same result was found when comparing the female patients to the male patients. This indicated that the older and female patients had a higher PNS response and a lower SNS response than the younger and male patients. It seemed that the older and female patients were confronted with the elevated PNS much more than the younger and male patients in controlling the stability of the ANS during the stroke event. This ANS pattern was also found in the circular and approximately rounded shapes of the Poincaré plots in the elderly and female patients, in which the PNS suppressed the SNS to the same level as indicating to the ischemic stroke. This evidence is supported by the mortality rate among ischemic stroke patients, as elderly females were at the highest risk [24]. Regarding the healthy participants, a previous study of short-term HRV found that there were no differences in SD1 and SD2 between males and females in both young and elderly groups [25].

Moreover, the results supported previous studies which found that the PNS and SNS were linked to HF and LF, respectively [26,27]. Additionally, the LF/HF ratio was linked to the balance between SNS and PNS [28]. The study suggested using the SD1 and PTT parameters for the PNS index, and the SD2 and SD2/SD1 parameters for the SNS index in ischemic strokes, after controlling for age and sex. Thus, the study recommended using the PNS parameters as a priority index for ischemic stroke.

The strengths of this study include that, to our knowledge, this is the first study that explores autonomic modulation through both linear and nonlinear HRV indices including PTT in Thai ischemic stroke patients. All assessments were noninvasive and performed by well-trained personnel using standardized protocol under standardized quality control systems. The presence of ischemic stroke in patients was verified by a full examination by the hospital, including the MRI scan. Nonetheless, there are some limitations to this study. First, HRV was performed using short-term recordings of 5 min. Generally, a longer duration for short-term recordings of 15–30 min is preferred. A more robust assessment of HRV data could have yielded a more rigorous standard of the measurement. Second, it should be emphasized that our HRV data represent daytime, which may not reflect the overall activities of PNS and SNS that originate from 24 h. Third, due to time limitations and access restrictions by the hospital, the assessment of cognitive performances was not possible and the level of missing data on the National Institutes of Health Stroke Scale (NIHSS) scores was high. Fourth, patients with diabetes, sleep apnea, and the use of cardio-selective drugs—all of which could have possible influence on their autonomic responses—were not excluded from the study. Fifth, the study did not collect personal or lifestyle information, such as smoking status, exercise patterns, occupation, or working hours, which could have added complementary information on autonomic control. This addresses the need for a large-scale prospective investigation of a 24-h Holter ECG for HRV measurements that include cognitive tests, NIHSS scores, sociodemographic information, clinical data, and related risk factors.

## 5. Conclusions

Among Thais who have experienced ischemic stroke, the PNS response was predominant in older and female patients while the SNS response was lower than in the younger and male patients. The older and female patients also showed in their Poincaré plots circular and approximately rounded shapes, respectively, with the SD1 radiuses close to the SD2 radiuses. These results suggested that there was an overactivity parasympathetic response to ischemic stroke in the elderly and female patients. Moreover, PNS response was positively associated with SD1 and PTT but negatively associated with SD2 and SD2/SD1, after controlling for age and sex, in all patients. Therefore, the study recommended using the SD1, SD2, SD2/SD1, and Poincaré plots including PTT as additional parameters in assessing autonomic modulation, particularly in the PNS, among ischemic stroke patients.

## Figures and Tables

**Figure 1 brainsci-11-00380-f001:**
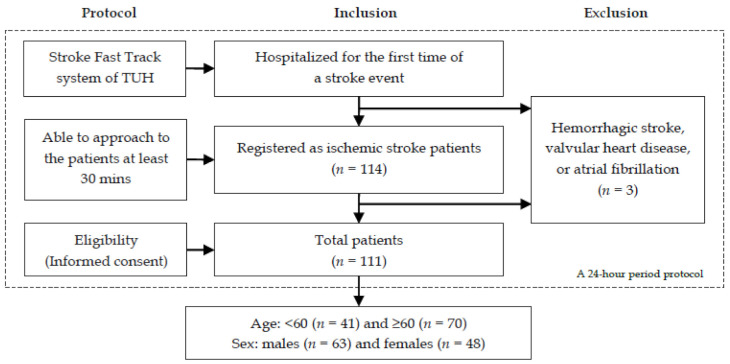
Flow diagram defining the study patients with a first ischemic stroke.

**Figure 2 brainsci-11-00380-f002:**
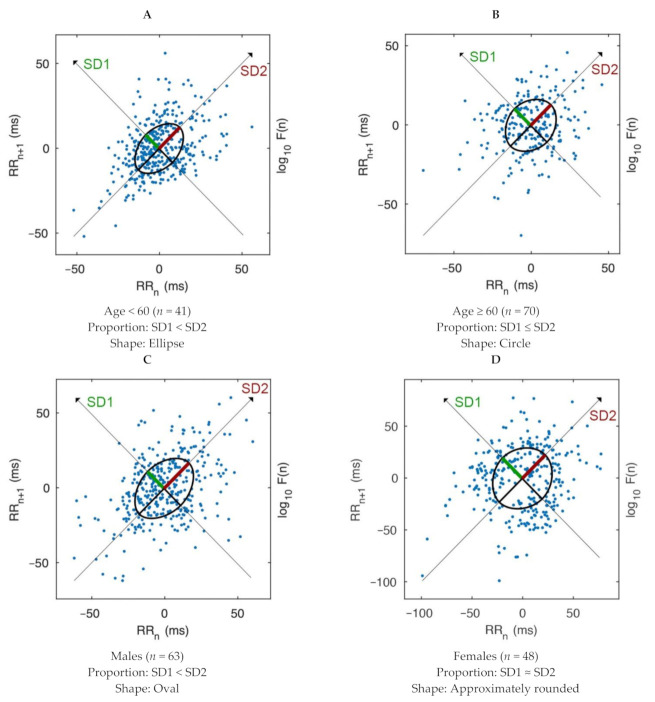
Comparison of Poincaré plots in all ischemic stroke patients by age and sex. Poincaré plots of patients under 60 years old (**A**). Poincaré plots of patients over 60 years old (**B**). Poincaré plots of male patients (**C**). Poincaré plots of female patients (**D**). Note: Poincaré plots were presented as representative of each group.

**Table 1 brainsci-11-00380-t001:** Means and standardization of all indices, ischemic stroke patients (*n* = 111).

Variable (Unit)	Total Patients
Age (years)	61.57 ± 12.64
Male (%)	56.80%
Autonomic indices	
PNS index	0.18 ± 2.00
SNS index	1.36 ± 1.91
HRV indices	
LF (n.u.)	46.63 ± 20.82
HF (n.u.)	52.85 ± 20.47
LF/HF ratio	1.45 ± 1.79
SD1 (%)	42.80 ± 9.48
SD2 (%)	57.17 ± 9.48
SD2/SD1	1.47 ± 0.62
PTT and BP indices	
PTT (ms)	0.24 ± 0.03
SBP (mmHg)	149.01 ± 20.03
DBP (mmHg)	82.95 ± 13.40
PP (mmHg)	66.05 ± 17.46
MAP (mmHg)	104.97 ± 13.63

**Table 2 brainsci-11-00380-t002:** Descriptive statistics and comparison results by age and sex in ischemic stroke patients.

Variable (Unit)	Age < 60 (*n* = 41)	Age ≥ 60 (*n* = 70)	*p* Value	Males (*n* = 63)	Females (*n* = 48)	*p* Value
Age (years)	49.00 (45.00–56.00)	69.00 (62.00–76.00)	<0.001 ^b^	62.06 ± 12.58	60.92 ± 12.82	0.638
Autonomic tones						
Parasympathetic tone			0.030 ^b^			0.826
Low	3 (7.30%)	1 (1.40%)		2 (3.20%)	2 (4.20%)	
Relatively low	11 (26.80%)	13 (18.60%)		13 (20.60%)	11 (22.90%)	
Average	21 (51.20%)	36 (51.40%)		33 (52.40%)	24 (50.00%)	
Relatively high	3 (7.30%)	7 (10.00%)		7 (11.10%)	3 (6.30%)	
High	3 (7.30%)	13 (18.60%)		8 (12.70%)	8 16.70(%)	
Sympathetic tone			0.281			0.905
Low	0	1 1.40(%)		0	1 (2.10%)	
Relatively low	2 (4.90%)	6 (8.60%)		5 (7.90%)	3 (6.30%)	
Average	17 (41.50%)	28 (40.00%)		25 (39.70%)	20 (41.70%)	
Relatively high	6 (14.60%)	17 (24.30%)		14 (22.20%)	9 (18.80%)	
High	16 39.00(%)	18 25.70(%)		19 (30.20%)	15 (31.30%)	
HRV indices						
LF (%)	43.00 ± 16.99	37.67 ± 16.22	0.104	42.35 ± 16.65	36.08 ± 16.09	0.048 ^a^
HF (%)	43.74 ± 20.70	49.88 ± 21.37	0.142	44.94 ± 19.95	51.11 ± 22.55	0.130
LF/HF ratio	1.16 (0.45–1.74)	0.86 (0.42–1.38)	0.167	1.16 (0.48–1.65)	0.70 (0.40–1.32)	0.106
SD1 (%)	40.17 ± 8.97	44.35 ± 9.49	0.024 ^a^	41.07 ± 8.34	45.08 ± 10.45	0.026 ^a^
SD2 (%)	59.83 ± 8.97	55.61 ± 9.49	0.023 ^a^	58.95 ± 8.33	54.84 ± 10.45	0.023 ^a^
SD2/SD1	1.55 (1.11–1.91)	1.26 (0.95–1.59)	0.014 ^b^	1.50 (1.15–1.85)	1.14 (0.94–1.59)	0.014 ^b^
PTT and BP indices						
PTT (ms)	0.24 (0.22–0.26)	0.24 (0.22–0.26)	0.971	0.24 (0.22–0.26)	0.24 (0.22–0.25)	0.520
SBP (mmHg)	149.00 (129.00–158.00)	150.00 (139.00–159.00)	0.308	149.00 (137.00–156.00)	150.00 (138.25–159.75)	0.555
DBP (mmHg)	83.00 (79.50–92.00)	82.00 (75.00–86.50)	0.041 ^b^	83.00 (77.00–90.00)	82.50 (76.25–89.50)	0.931
PP (mmHg)	60.00 (48.50–67.00)	67.00 (58.75–78.00)	0.004 ^b^	67.00 (56.00–72.00)	67.00 (55.50–74.75)	0.903
MAP (mmHg)	106.71 ± 14.84	103.96 ± 12.87	0.307	104.41 ± 12.95	105.71 ± 14.57	0.622

Note: Results were presented as median and interquartile ranges (25–75°) or mean ± standard deviation or number (%). Significant at *p* < 0.05 with ^a^ independent *t*-test or ^b^ Mann–Whitney *U* test.

**Table 3 brainsci-11-00380-t003:** A nonparametric partial correlation in all ischemic stroke patients with age and sex adjusted.

Variable (Unit)	PNS Index		SNS Index	
	Correlation (*r*)	*p* Value	Correlation (*r*)	*p* Value
HRV indices				
LF (n.u.)	−0.38	<0.001 *	0.27	0.004 *
HF (n.u.)	0.38	<0.001 *	−0.28	<0.001 *
LF/HF ratio	−0.37	<0.001 *	0.29	0.002 *
SD1 (%)	0.47	<0.001 *	−0.39	<0.001 *
SD2 (%)	−0.47	<0.001 *	0.39	<0.001 *
SD2/SD1	−0.47	<0.001 *	0.40	<0.001 *
PTT and BP indices				
PTT (ms)	0.29	0.002 *	−0.21	0.026 *
SBP (mmHg)	−0.09	0.372	0.09	0.373
DBP (mmHg)	−0.24	0.012 *	0.28	0.003 *
PP (mmHg)	0.04	0.679	−0.08	0.404
MAP (mmHg)	−0.17	0.086	0.18	0.066

Note: * denotes significance level at *p* < 0.05.

## Data Availability

The data are not publicly available due to the restrictions imposed by data providers and the ethical approval that governs the fieldwork. Access to data would only be granted upon request. For further inquiries, please contact the first author at warawoot.c@chula.ac.th.

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
