# Peer review of "Variation of Autonomic Nervous System Function by Age and Gender in Thai Ischemic Stroke Patients"

_brainsci, 2021, doi:10.3390/brainsci11030380_

Round 1

Reviewer 1 Report

This manuscript explore the role of the autonomic nervous system (ANS) through nonlinear heart rate variability (HRV) and pulse transit time (PTT)  in Ischemic stroke.

I found this topic interesting and the study was thoroughly and well conducted. I only have some concerns relating to the implication of results.  While the clinical implications are obvious, using even nonlinear indices as markers of a complex cardiovascular system, some indirect implications are missing, such as those on cognitive function (it is known that HRV is associated with cognitive impairment. for reviews see: Forte, G., Favieri, F., & Casagrande, M. (2019). Heart rate variability and cognitive function: a systematic review. Frontiers in neuroscience13, 710.).

Author Response

Dear Reviewer #1,

RE: Manuscript #: brainsci-1108091

On behalf of all co-authors, I would like to thank for the comments and suggestions, all of which have been accounted for in the revision of the manuscript. Please find below our responses to the reviewer’s comments. Note that all responses have been incorporated in the texts of the revised manuscript enclosed with this letter.

Reviewer #1,

Comment #1. I found this topic interesting and the study was thoroughly and well conducted. I only have some concerns relating to the implication of results. While the clinical implications are obvious, using even nonlinear indices as markers of a complex cardiovascular system, some indirect implications are missing, such as those on cognitive function (it is known that HRV is associated with cognitive impairment. for reviews see: Forte, G., Favieri, F., & Casagrande, M. (2019). Heart rate variability and cognitive function: a systematic review. Frontiers in neuroscience, 13, 710.).

Response: Due to time limitations and access restrictions by the hospital, it was not possible to perform cognitive tests. This has been added as a limitation of this study in Section 4 (Discussion) of the revised manuscript.

General revision: All keywords have been changed to lowercase letters; “gender” has been changed to “sex” where appropriate; spaces after < and ≥ have been removed; Funding, Acknowledgements, Author contributions, citations and References have been corrected; Institutional review board statement, Informed consent statement, and Data availability statement have been added as new sections in the revised manuscript.

I truly believe that the quality of the manuscript has been much improved after the modifications according to the reviewers’ recommendations and wish that it would be now deemed worthy for publication in Brain Sciences.

Thank you in advance for your consideration of our revised manuscript. We look forward to your reply.

Sincerely yours,

Reviewer 2 Report

Chuangchai et al. described the autonomic nervous system modifications in a cohort of Thai patients with ischemic stroke by means of non-linear parameters of heart rate variability and pulse transit time. They found an increased parasympathetic tone in elderly patients and in woman compared to younger patients and male. The paper is of interest and evaluate an important aspect of the management of ischemic stroke.

There are some major concerns that should be addressed as follow:

  1. The patients should include a limitations section where they described the limits of the study. For example, patients affected by diabetes or sleep apnea where not excluded. Both OSA and diabetes are well known modifying factors of HRV. Similarly, the use of cardio-selective drugs was not an exclusion criteria and this should be listed in the limitations section too.
  2. It is not clear in which phase of ischemic stroke (acute? Sub-acute?) the ECG was recorded. This should be clarified in the study.
  3. In the method’s section should be listed the variables collected and analyzed
  4. Clinical characteristics of study group are missing. ANS modifications have a different impact in the posterior circulation stroke or anterior circulation stroke. Data regarding the stroke location should be added in the manuscript. Moreover, ANS are more evident in right stroke, particularly in insular lesions. (Refer to: Autonomic Nervous System Modifications During Wakefulness and Sleep in a Cohort of Patients with Acute Ischemic Stroke. Brunetti V, et al. J Stroke Cerebrovasc Dis. 2019 Jun;28(6):1455-1462. doi:10.1016/j.jstrokecerebrovasdis.2019.03.023.). Adjusting findings for stroke location would strengthen findings of the current paper. Data of stroke severity (i.e. NIHSS) are mandatory to interpret the results.
  5. What are autonomic indices expressed in the table 1? This is not clear and how they are calculated should be clarified in the method section.
  6. In the discussion the Authors state that “The results confirmed that ANS dysregulation among ischemic stroke patients varied with age and gender.” The study doesn’t have a control group; from this point of view the mean findings of the current manuscript could be independent by the presence of ischemic stroke but simply related to physiological differences between age and sex in ANS modulation. This point should be reformulate
  7. The Authors state that “The presence of ischemic stroke in patients was verified by a full examination by the hospital.”. Do the Authors mean that was confirmed by neuroimaging? This should be clarified.

Author Response

Dear Reviewer #2,

RE: Manuscript #: brainsci-1108091

On behalf of all co-authors, I would like to thank the reviewer for the comments and suggestions, all of which have been accounted for in the revision of the manuscript. Please find below our responses to each of the reviewer’s comments. Note that all responses have been incorporated in the texts of the revised manuscript enclosed with this letter.

Comment #1. The patients should include a limitations section where they described the limits of the study. For example, patients affected by diabetes or sleep apnea where not excluded. Both OSA and diabetes are well known modifying factors of HRV. Similarly, the use of cardio- selective drugs was not an exclusion criteria and this should be listed in the limitations section too.

Response: All suggestions by the reviewer have been included as the limitations of this study in Section 4 (Discussion).

Comment #2. It is not clear in which phase of ischemic stroke (acute? Sub-acute?) the ECG was recorded. This should be clarified in the study.

Response: HRV data were recorded in an acute phase of ischemic stroke. Section 2 (Measurements) has been revised to clarify this point.

Comment #3. In the method’s section should be listed the variables collected and analyzed.

Response: All variables have been listed in Section 2.2 (Measurements) and further described in Section 2.3 (Statistical analysis).

Comment #4. Clinical characteristics of study group are missing. ANS modifications have a different impact in the posterior circulation stroke or anterior circulation stroke. Data regarding the stroke location should be added in the manuscript. Moreover, ANS are more evident in right stroke, particularly in insular lesions. (Refer to: Autonomic Nervous System Modifications During Wakefulness and Sleep in a Cohort of Patients with Acute Ischemic Stroke. Brunetti V, et al. J Stroke Cerebrovasc Dis. 2019 Jun;28(6):1455-1462. doi:10.1016/j.jstrokecerebrovasdis.2019.03.023.). Adjusting findings for stroke location would strengthen findings of the current paper. Data of stroke severity (i.e. NIHSS) are mandatory to interpret the results.

Response: All patients in this study were diagnosed with right middle cerebral artery (R-MCA) infarction, except one that had left MCA infarction. No insular lesions were detected. Section 2.1 (Participants) has been revised to include this information. In addition, as the level of missing data in the NIHSS scores was high, we decided not to show the results. This has been mentioned as a limitation of our study in the last paragraph of Section 4 (Discussion).

Comment #5. What are autonomic indices expressed in the table 1? This is not clear and how they are calculated should be clarified in the method section.

Response: The ANS results were obtained using Kubios HRV software based on the mean RR, RMSSD, and SD1 parameters for the PNS index, and on the mean HR, stress index, and SD2 parameters for the SNS index, respectively. The formulae used to obtain these indices by the software are, however, not publicly available. Section 2.3 (Statistical analysis) has been revised to include this clarification. Also, further information on the parasympathetic nervous system (PNS) and the sympathetic nervous system (SNS), as well as their links to different HRV parameters, has been provided in Section 1 (Introduction).

Comment #6. In the discussion the Authors state that “The results confirmed that ANS dysregulation among ischemic stroke patients varied with age and gender.” The study doesn’t have a control group; from this point of view the mean findings of the current manuscript could be independent by the presence of ischemic stroke but simply related to physiological differences between age and sex in ANS modulation. This point should be reformulate.

Response: The reviewer’s point has been incorporated in the first paragraph of Section 4 (Discussion) in the revised manuscript.

Comment #7. The Authors state that “The presence of ischemic stroke in patients was verified by a full examination by the hospital.”. Do the Authors mean that was confirmed by neuroimaging? This should be clarified.

Response: MRI was used as part of the examination for all patients in this study. This has been stated in Section 2.1 (Participants) as well as in Section 4 (Discussion) of the revised manuscript.

General revision: All keywords have been changed to lowercase letters; “gender” has been changed to “sex” where appropriate; spaces after < and ≥ have been removed; Funding, Acknowledgements, Author contributions, citations and References have been corrected; Institutional review board statement, Informed consent statement, and Data availability statement have been added as new sections in the revised manuscript.

I truly believe that the quality of the manuscript has been much improved after the modifications according to the reviewers’ recommendations and wish that it would be now deemed worthy for publication in Brain Sciences.

Thank you in advance for your consideration of our revised manuscript. We look forward to your reply.

Sincerely yours,
